# Great patients embed alike: contrastive learning for sample representation from single-cell data

## Abstract

Single-cell transcriptomics has revolutionized cellular biology by measuring gene activity in thousands of cells per donor, giving insights into cellular processes in normal tissue and early-stage disease. Sample representation methods encode all cells from one patient as a single patient vector, enabling applications of single-cell data for health state classification, prediction of future clinical status, and patient stratification. However, current single-cell datasets have fewer than a thousand samples, making it challenging for the models to learn generalisable and robust sample representations. To overcome this limitation, we suggest learning sample representations in a self-supervised way, relying on set representation invariance to subsampling. We develop SampleCLR, a contrastive learning method, which can be extended to supervised task prediction by the multiple instance learning framework[1]. We show that SampleCLR outperforms unsupervised methods when trained in a self-supervised way, and reaches state-of-the-art quality of sample representation when fine-tuned on a supervised task, despite having orders of magnitude fewer parameters than other methods. We further demonstrate that SampleCLR is interpretable by design via cell importance module and learns signatures of COVID-19 severity. We envision SampleCLR to pave the way for diagnostic applications from single-cell data.

## 1 Introduction

Single-cell transcriptomics measures gene expression in thousands of cells per sample in high throughput (Gulati et al. (2025)). This has been used to find new cell types in human tissue (Montoro et al. (2018)) and uncover disease processes at early stages (Kim et al. (2021)). As single-cell datasets are growing in the number of cells and donors profiled (Hrovatin et al. (2025)), so-called sample representation methods are emerging that represent the entire sample as a single vector (Shitov et al. (2025)). These sample representations can be used to predict health status, stratify patients, and find molecular features that explain sample-level phenotypes, promising translation of single-cell transcriptomics to clinical applications (Joodaki et al. (2024); Boiarsky et al. (2025); Liu et al. (2025)). However, existing methods do not fully leverage data, missing the clinically relevant information in the representation, and are often outperformed by simple baselines (Shitov et al. (2025)).

Currently, two groups of methods exist for sample representation from single-cell transcriptomics data. Unsupervised approaches use only cellular features (e.g., gene expression) to infer the similarity of samples in the dataset (Joodaki et al. (2024); Wang et al. (2024); De Donno et al. (2023); Argelaguet et al. (2020); Boyeau et al. (2024)). While unsupervised learning can successfully capture strong sources of biological variation, such as disease severity, it is also susceptible to strong technical variation, such as batch effects, and does not allow researchers to focus on particular effects of interest that may be subtle. For example, if the data contains information to successfully prognose future disease outcome, it can be hidden behind stronger effects related to current disease severity. Supervised methods can remedy this limitation by creating sample representations that are well-suited for any particular prediction task (Boiarsky et al. (2025); Liu et al. (2025); Querfurth et al. (2024); Litinetskaya et al. (2024); Engelmann et al. (2024)).

---

[1]The code is available at https://github.com/sampleclr-iclr2026/SampleCLR

The few currently available supervised sample representation methods, however, focus only on one supervised task. This can result in simplified sample representations, missing the inherent diversity of the samples with the same label. Moreover, supervised methods can suffer from incorrect labels and are not able to handle missing labels. In contrast, a self-supervised learning paradigm can effectively leverage the structure of the data without labels, even outperforming supervised methods (Chen et al. (2020); Richter et al. (2025)). But the application of self-supervised learning to sample representation from single-cell data has remained limited.

In this paper, we introduce SampleCLR, the first framework for self-supervised sample representation from single-cell data using contrastive learning. We derive SampleCLR from the natural symmetry of single-cell experiments to cell subsampling (Section 3.4). We show that SampleCLR provides high-quality sample embeddings applicable to a wide range of downstream tasks, whether trained in a self-supervised way or with supervised fine-tuning (Section 5.1). Finally, we show how the cell importance module of SampleCLR enables interpretability of the sample embeddings on a cellular and molecular level, allowing researchers to derive biological insights (Section 5.2). Our work contributes to the set representation learning (Zaheer et al. (2017)) and paves the way for clinical applications of single-cell transcriptomics.

Our main contributions are:

1. The application of contrastive learning to sample representation via subsampling augmentation;

2. SampleCLR – method for sample representation from single-cell data by connecting contrastive and multiple-instance learning;

3. Empirical evidence that self-supervised learning helps to achieve state-of-the-art quality of sample representation;

4. Empirical evidence that SampleCLR learns biologically meaningful signals that are relevant to apply the method to clinical challenges.

## 2 RELATED WORK

### 2.1 SAMPLE REPRESENTATION

Sample representation methods embed a set of cells coming from a donor into a latent vector or provide a distance matrix between samples in the data (Joodaki et al. (2024); Wang et al. (2024); De Donno et al. (2023); Argelaguet et al. (2020); Boyeau et al. (2024)). Unsupervised methods only take cell information into account and do not use sample metadata. Among unsupervised sample representation methods, a density-based approach, GloScope (Wang et al. (2024)), stands out as it captures the most information from the data according to an independent benchmark (Shitov et al. (2025)). GloScope calculates a symmetrised Kullback-Leibler divergence between estimated cell densities for each pair of donors.

Supervised methods are specifically trained to solve a downstream task on a sample level, such as disease classification. For set data, the multiple-instance learning (MIL) approach is commonly used (Carbonneau et al. (2018)). Engelmann et al. (2024) suggested MixMIL – a framework that integrates generalised linear models and MIL for supervised prediction of sample labels. MixMIL uses a shallow neural network to learn cell importance weights for aggregation. Litinetskaya et al. (2024) suggested MultiMIL, another MIL-based approach for sample-level prediction with gated attention and simultaneous learning of cell representations via reconstruction loss. Querfurth et al. (2024) suggested mcBERT, a transformer-based model with self-supervised pretraining of a "student" model with masked cell features to match the outputs of a "teacher" model having access to all the features (Baevski et al. (2022)). Despite having a self-supervised pre-training, the authors notice that it does not provide high-quality sample representations, and supervised fine-tuning is needed. The mcBERT is further trained in a supervised mode by supervised contrastive learning (Khosla et al. (2020)). Liu et al. (2025) developed PaSCient, a neural network consisting of an aggregator with learnable cell weights and a classifier to predict disease labels.

Currently, unsupervised and supervised sample representation methods are disconnected, and there is a limited application of self-supervised learning methods. In this work, we aim to close this

gap and provide a consistent framework for learning high-quality sample representations both in a self-supervised and in a supervised way.

## 2.2 CONTRASTIVE LEARNING

Contrastive learning (CL) is a powerful technique for learning representations of data points by pulling together embeddings of similar data points (positive pairs) while pushing them apart from the embeddings of other non-similar objects (negative pairs). Chen et al. (2020) showed that contrastive learning can produce high-quality representations of images, outperforming supervised models in classification. Furthermore, CL can provide a meaningful substructure of the data as demonstrated in images (Böhm et al. (2022)) and text (González-Márquez et al. (2024)). Sobal et al. (2024) introduced $\mathbb{X}$-sample contrastive loss, enabling training representations given a prior data similarity matrix. In imaging histopathology, CL was successfully combined with MIL for learning whole slide image representation (Tavolara et al. (2022)). Despite an increasing interest in CL for single-cell data (Ji et al. (2025); Zhao et al. (2025); Richter et al. (2025)), its application for sample representation has been largely unexplored.

## 2.3 BENCHMARKING METRICS

Shitov et al. (2025) provides a benchmark of unsupervised methods for sample representation from single-cell data called SPARE. The authors introduced 4 metrics to evaluate the quality of sample embeddings:

- **Information retention**, which measures how well sample metadata can be predicted from a local neighborhood of a sample. This is done by predicting clinically relevant metadata by k nearest neighbours in the sample representaion. The score is the average prediction quality measured by $F_1$-macro score for categorical features (such as disease status) and Spearman correlation for continuous features (such as disease severity).

- **Batch mixing**, which validates that technical effects are not co-localised in the embedding. This is measured in the same way as information retention, but the score is inverted so that 0 means perfect separation and 1 represents perfect batch mixing.

- **Trajectory conservation**, which evaluates a global structure of sample embedding by correlating pseudotime with ground-truth trajectory, such as disease severity or age;

- **Replicate robustness**, which measures if replicates of the same biological sample have similar embeddings.

## 3 BACKGROUND AND ASSUMPTIONS

### 3.1 SINGLE-CELL TRANSCRIPTOMICS

A single-cell transcriptomics dataset $\mathbf{X}$ is a collection of $n$ cells. For each cell, a vector of gene expression is measured. Initially, it consists of tens of thousands of features (i.e., genes), which are filtered and embedded into a lower-dimensional space during preprocessing (Heumos et al. (2023)). Each cell can then be thought of as a vector $\mathbf{X}_i$ of $d$ features, so $\mathbf{X} \in \mathbb{R}^{n \times d}$. The dimensionality $d$ typically ranges from 10 to 50 when variational autoencoders (Kingma & Welling (2022); Xu et al. (2021); De Donno et al. (2023)) or principal component analysis (PCA) (Pearson (1901)) is applied. Each cell in an experiment is associated with a vector of categorical labels $T$, $T_i \in \{1, 2, \ldots, K\}$ called cell types. In typical experiments, $K$ is of the order of tens. Additionally, in multi-sample experiments, cells are associated with sample labels $L$, $L_i \in \{1, 2, \ldots, M\}$. In recent years, public datasets have reached up to a thousand samples (Hrovatin et al. (2025)), while collections of single-cell datasets, called atlases, integrate datasets to build resources with thousands of samples. Finally, each sample has an associated metadata table, which can contain categorical covariates, such as disease label, ordered covariates, such as disease severity, and continuous features, such as age. Following Shitov et al. (2025), we split metadata covariates into clinically relevant, which must be preserved in sample representation, and technical, which should not affect the representation.

## 3.2 Strengths and limitations of the pseudobulk baseline

A common baseline for sample representation is "pseudobulk" – an average cellular profile of a sample with $n_i$ cells:

$$f_{\text{bulk}}(X_i) = \frac{1}{n_i} \sum_{k=1}^{n_i} X_i^k \tag{1}$$

Despite its simplicity, pseudobulking proves to be a very powerful baseline, outperforming more complex methods (Shitov et al. (2025)). However, pseudobulking has a **critical limitation**: it is not able to distinguish samples with identical means but different distributions. Consider two samples with identical means and cell type proportions:

- Sample A: All cells express an inflammatory marker at level 5
- Sample B: Half the cells express at level 0, half at level 10

Both have an average value of 5, but Sample B shows a bimodal distribution while Sample A shows uniform expression. Pseudobulking cannot distinguish these biologically different scenarios.

## 3.3 Building on top of pseudobulk power: learned importance weighting

Pseudobulking can be seen as a weighted sum of cells with identical weights $\frac{1}{n_i}$. This contradicts the intuition: cells should have different weights depending on how much they contribute to the difference between samples. We suggest letting a model learn the cell importance weights. In the example above, a model could learn that cells with high expression of a gene enable distinguishing the samples and therefore should get higher weights, while other cells can be de-prioritised. By learning cell importance, a model can overcome pseudobulk limitations and provide interpretability of which cells contribute to sample representation.

## 3.4 Sample representation is invariant to cell subsampling

A central assumption in single-cell genomics is that a subset of cells that is sampled from donor tissue can represent the whole biological system (for example, a patient's lung) well. Formally, let $R$ be an ideal representation method and let $S_p$ denote an operation that randomly subsets a fraction $p$ of objects in a set. Then, if ♟ is the set of all cells in a given tissue of a donor, a single-cell transcriptomics experiment is $S_p(♟)$. The assumption is that if $p$ is sufficiently large, then

$$\|R(S_p(♟)) - R(♟)\| < \varepsilon \tag{2}$$

for some norm $\|\cdot\|$ and a small $\varepsilon \in \mathbb{R}$. In other words, a subset of cells captures all the essential information about the tissue of a donor. It is then natural to assume that when the number of measured cells is big enough, further subsetting can be performed without losing information. If we then split a single-cell sample $S_p(♟)$ into two non-overlapping subsamples of cells $S^1$ and $S^2$, we obtain

$$\|R(S^1) - R(♟)\| < \varepsilon \quad \text{and} \quad \|R(S^2) - R(♟)\| < \varepsilon. \tag{3}$$

Thus, representations of different subsamples $R(S^1)$ and $R(S^2)$ must be similar to each other. We further assume that they must also not be similar to representations of other samples with different biological conditions. We can model $R$ as a neural network and train it in a contrastive way to satisfy these assumptions.

# 4 SampleCLR for self-supervised sample representation for single-cell data

We suggest SampleCLR – a framework for learning sample representations from single-cell data in a self-supervised way by training a neural network to maximise similarities of different cell subsamples from the same donor (Figure 1). On every training step, 2 sets of cells are sampled from each donor in a minibatch. They are passed through SampleCLR to get representations. The neural

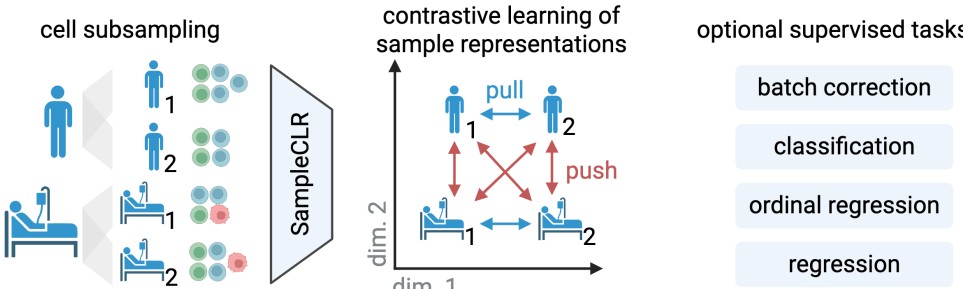

Figure 1: SampleCLR approach for sample representation from single-cell data. Cells from samples are randomly subsampled to form positive and negative pairs, and sample representations are learned in a contrastive way with optional supervised tasks on top.

networks in SampleCLR are then updated by minimising $\mathbb{X}$-Sample Contrastive Loss (Sobal et al. (2024)). By default, it is equivalent to InfoNCE loss (van den Oord et al. (2018)), considering subsamples from the same sample a positive pair and all the other subsamples as a negative pair. However, $\mathbb{X}$-Sample Contrastive Loss enables the input of a prior sample similarity matrix. In our experiments, we use a sample similarity matrix of GloScope as a prior, leveraging the power of this method.

To obtain sample representation $z_i$ for a given set of cells $X_i$, SampleCLR uses the cell importance module $w$ and encoder $f$ (Figure 2). To enable direct interpretability of the model, we add the cell importance module, which assigns a non-negative weight for each cell. Intuitively, this can be viewed as the attention that the model pays to each cell. To allow the model to learn different patterns in the dataset, we define $H$ cell importance heads, each providing its own vector of cell importances. The cells are then aggregated by a weighted sum according to their learned importances. Aggregated representations from each importance head are concatenated to form the vector $a_i$. Encoder neural network takes $a_i$ to form the final representation $z_i$.

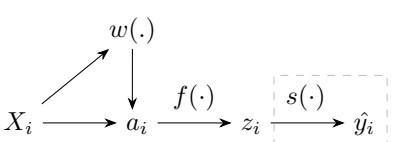

Figure 2: SampleCLR model scheme. $X_i$ – a set of cells from sample $i$, $w$ – cell importance module, $a_i$ – aggregated representation, $f$ – encoder, $z_i$ – sample representation, $s$ – supervised module, $\hat{y}_i$ – supervised task prediction. Optional elements are surrounded by a dashed box.

Optionally, supervised tasks can be added on top of the SampleCLR to predict a sample-level covariate of interest and let it shape the sample representation space. We do it by adding fully-connected neural networks $s$ to $z_i$. SampleCLR supports classification of categorical covariates, ordinal regression of ranked features, and regression for continuous sample variables. For the formal description of SampleCLR, see Algorithm 1.

SampleCLR supports classification, regression and ordinal regression as supervised tasks. In each case, $s$ is a fully-connected neural network. For classification task, it has output dimensionality $C$ according to the number of classes. Cross entropy is used as a loss function for predicted logits $\hat{y}_i$ and one-hot-encoded target $y$:

$$L_i^{class} = -\sum_{c=1}^{C} \log \frac{\exp(\hat{y}_{i,c})}{\sum_{k=1}^{C} \exp(\hat{y}_{i,k})} y_{i,c} \qquad (4)$$

Ordinal regression is applied to ranked features: classes with order. Supervised head in this case has output dimension $C-1$. $k$-th output element indicates whether target is greater than $k$. Binary cross entropy is then used as loss function for true and predicted target values:

$$L_i^{ord-reg} = - \sum_{k=1}^{C-1} [y_{i,k} \log(\hat{p}_{i,k}) + (1 - y_{i,k}) \log(1 - \hat{p}_{i,k})] \tag{5}$$

Where $\hat{p}_{i,k} = \sigma(\hat{y}_{i,k})$ are predicted probabilities that the target $y$ is greater than $k$.

Finally, for regression, the supervised head outputs one number and mean squared error loss is calculated between true and predicted target:

$$L_i^{regr} = (\hat{y}_i - y_i)^2 \,, \tag{6}$$

Weights of encoder network $f$ and supervised head $s$ are updated to minimise the corresponding loss. SampleCLR also supports multi-task learning by predicting several metadata covariates at once.

---

**Algorithm 1** SampleCLR self-supervised training step

---

**Input:** size $N$;
  temperature $\tau$;
  minibatch $\{X_k, y_k\}_{k=1}^N$;
  structure of $w$ (cell importance module);
  structure of $f$ (encoder);
  prior sample similarity graph $S$
**for** *sampled minibatch* $\{X_k\}_{k=1}^N$ **do**
  **for** $k \in \{1, \ldots, N\}$ **do**
    draw two subsamples of $X_k$: $\tilde{X}_{2k-1}, \tilde{X}_{2k}$
    **for** $v \in \{2k-1, 2k\}$ **do**
      $I_h \leftarrow \{w_h(\tilde{X}_v)\}_{h=1}^H$ ;        // cell importances from $H$ heads
      $a_v \leftarrow Concat(\{\sum \tilde{X}_v \odot I_h\}_{h=1}^H)$ ;   // weighted aggregation of cells
      $z_v \leftarrow f(a_v)$ ;                    // sample representation
  **for** $i \in \{1, \ldots, 2N\}$ *and* $j \in \{1, \ldots, 2N\}$ **do**
    $s_{i,j} \leftarrow \dfrac{z_i^\top z_j}{\|z_i\| \|z_j\|}$ ;                    // pairwise similarity
  $p(i,j) \leftarrow \dfrac{\exp(s_{i,j}/\tau)}{\sum_{k=1}^{2N} \mathbf{1}_{[k \neq i]} \exp(s_{i,k}/\tau)}$
  $\mathcal{L}_{\mathbb{X}\text{-CLR}} \leftarrow \frac{1}{2N} \sum_{k=1}^{2N} \text{CE}(S_i, p_i)$ ;   // Cross-entropy to prior similarities
  update $w, f$ to minimise $\mathcal{L}_{\mathbb{X}\text{-CLR}}$
  **if** *supervised* **then**
    $\hat{s}_i \leftarrow s(a_i)$
    $\mathcal{L}_{sup} \leftarrow \text{task\_loss}(\hat{s}_i, s_i)$
    Update $s, w, f$ to minimise $\mathcal{L}_{sup}$

---

To get the final sample representations from a trained network, we use a fixed number of cells for each dataset. We use 1000 cells with the lowest eigenvector centrality values based on the nearest neighbour cell graph (see Appendix A.5).

## 5 EXPERIMENTS

### 5.1 BENCHMARKING

We benchmarked SampleCLR against a wide range of unsupervised and supervised methods (Table 1) on 4 datasets from blood and lung tissues. We assessed the sample representation methods on their ability to capture biomedically relevant signals in the embeddings, mimicking a potential clinical application scenario. Specifically, we looked for preservation of COVID-19 severity in two blood

datasets, an aging signature in a further blood dataset, as well as anatomical location preservation and disease classification in the human lung cell atlas (Appendix). We apply SPARE metrics (Section 2.3) to assess that similar samples have similar metadata, the global structure of embeddings reflects the ground truth trajectory, and that batch effects do not dominate the representation.

SampleCLR trained in a self-supervised way reaches the aggregated score of the best-performing unsupervised method, GloScope, and outperforms most of the supervised methods besides PaSCient. When SampleCLR is additionally fine-tuned using a supervised learning module, SampleCLR outperforms all the methods of comparison. Notably, it does so by having orders of magnitude fewer parameters. SampleCLR trained for COVID-19 datasets has only 10 thousand parameters, and the version used for HLCA and onek1k datasets contains 370 thousand trainable parameters. In contrast, the second-best method, PaSCient, has 4,9 million parameters, and mcBERT has 26 million parameters, taking significantly more resources for training. High quality of sample representations with smaller models indicates that SampleCLR self-supervised pre-training efficiently uses information in the data.

The metric breakdown shows that SampleCLR excels in trajectory preservation and batch mixing, while being outperformed by GloScope in local information retention (Table 1). This can be related to the fact that GloScope uses all samples in the dataset to build sample representation, while SampleCLR is not trained on part of the dataset used as a validation set.

We further tested the effect of using the prior similarity graph on the performance of SampleCLR by using GloScope representation as a prior. Interestingly, in an unsupervised scenario, SampleCLR does not benefit from using a prior similarity graph and indeed performs better without it (total score 0.53 without a prior and 0.51 with a prior). However, when the model is fine-tuned using the supervised learning module, using a prior similarity graph makes representations more informative and stable (total score $0.55 \pm 0.071$ without a prior and $0.58 \pm 0.033$ with a prior). We presume that the prior similarity graph acts as additional regularisation, forcing the model to keep representations close to the prior while enriching them by solving a supervised task.

Table 1: Benchmarking results across evaluation metrics. The mean and standard deviations are shown across benchmarking datasets. * marks Sample CLR (our model)

| | Information Retention | Batch Mixing | Trajectory Preservation | Total Score |
|---|---|---|---|---|
| **Supervised methods** | | | | |
| SampleCLR supervised + prior* | $0.47 \pm 0.083$ | $\mathbf{0.61} \pm 0.167$ | $\mathbf{0.68} \pm 0.110$ | $\mathbf{0.58} \pm 0.033$ |
| SampleCLR supervised* | $\mathbf{0.48} \pm 0.109$ | $0.60 \pm 0.146$ | $0.60 \pm 0.241$ | $0.55 \pm 0.071$ |
| PaSCient | $0.45 \pm 0.240$ | $0.52 \pm 0.164$ | $0.65 \pm 0.111$ | $0.54 \pm 0.101$ |
| MultiMIL | $0.45 \pm 0.087$ | $0.43 \pm 0.117$ | $0.60 \pm 0.174$ | $0.51 \pm 0.072$ |
| mcBERT | $0.41 \pm 0.136$ | $0.50 \pm 0.237$ | $0.61 \pm 0.083$ | $0.51 \pm 0.040$ |
| MixMIL | $0.47 \pm 0.188$ | $0.35 \pm 0.159$ | $0.32 \pm 0.404$ | $0.38 \pm 0.207$ |
| **Unsupervised methods** | | | | |
| GloScope | $\mathbf{0.51} \pm 0.137$ | $0.42 \pm 0.160$ | $\mathbf{0.60} \pm 0.170$ | $\mathbf{0.53} \pm 0.064$ |
| SampleCLR unsupervised* | $0.45 \pm 0.036$ | $\mathbf{0.57} \pm 0.115$ | $0.58 \pm 0.190$ | $\mathbf{0.53} \pm 0.054$ |
| SampleCLR unsupervised + prior* | $0.42 \pm 0.070$ | $0.56 \pm 0.115$ | $0.58 \pm 0.190$ | $0.51 \pm 0.054$ |
| Pseudobulk | $0.32 \pm 0.020$ | $0.55 \pm 0.096$ | $0.53 \pm 0.209$ | $0.45 \pm 0.127$ |
| Cell type composition | $0.33 \pm 0.179$ | $0.54 \pm 1.147$ | $0.25 \pm 0.202$ | $0.34 \pm 0.152$ |

## 5.2 BIOLOGICAL INTERPRETATION OF CELL WEIGHTS

The cell aggregator module of SampleCLR enables interpretation of the sample representations on a cellular and molecular scale by highlighting which cells are taken into account when forming the representations. To demonstrate how this can impact biological discovery, we analysed the importance weights for cells in a sample representation of the COVID-19 COMBAT dataset. We observed that importance weights appear to be structured by cell type and different aggregator heads put emphasis on different cell types (Figure 3a), despite cell type information not being provided to

the model. Platelets and erythrocytes, which received the highest importance values, are known to change proportions depending on COVID-19 severity (Stephenson et al. (2021)), highlighting that the model learned robust patterns associated with disease.

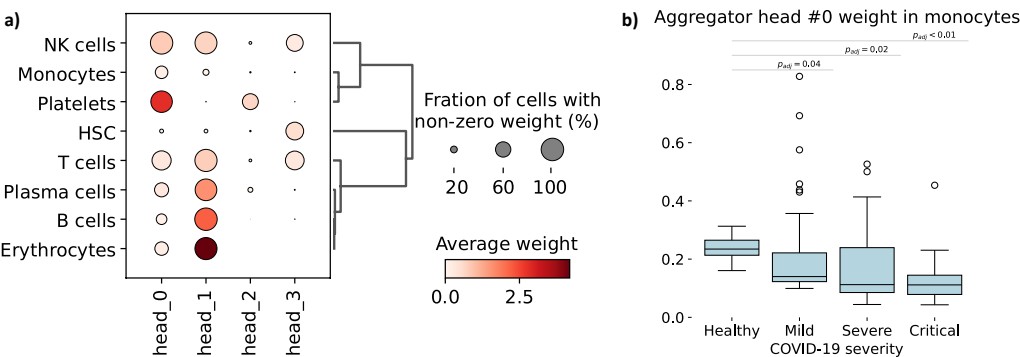

Figure 3: SampleCLR enables interpretability by design and learns meaningful biological patterns in the data. **a)** Cell weights visualisation for supervised SampleCLR model trained on COMBAT COVID-19 dataset. The x-axis shows 4 aggregation heads of the model. The y-axis shows different cell types in the dataset. **b)** Per sample distribution of cell weights from aggregator head #0 in monocytes.

To further understand if the weights of the aggregator heads learn more intricate patterns within cell types, we tested the association of cell weights from head 0 to COVID-19 severity in monocytes (Figure 3b). We see that weights negatively correlate with disease severity, clearly distinguishing healthy donors from COVID-19 patients. To test whether cell weights highlight specific gene programs within cell types, we computed the activity of the MSigDB Hallmark 2020 gene sets (Liberzon et al. (2015)). We then calculated the Spearman correlation between cell weights from SampleCLR and each pathway activity score. We find that cell weights of aggregation head 0 in classical and non-classical monocytes correlate the most with Interferon Alpha Response ($\rho$ = 0.48 and 0.42, respectively), a known marker of COVID-19 severity (COvid-19 Multi-omics Blood ATlas (COMBAT) Consortium (2022)). In summary, we see that SampleCLR is interpretable by design and learns meaningful biological patterns in the data, providing biologists with a tool to better understand diseases.

## 6 DISCUSSION

We presented SampleCLR, a self-supervised method for sample representation learning from single-cell data. Our method is grounded on the subsampling symmetry of set data and performs strongly in practice. We have shown that SampleCLR provides high-quality sample representations across different tissues and tasks and enables interpretability for biological conclusions. Limitations of the current work include training SampleCLR on one dataset at a time. In future work, it can be applied to large integrated collections of datasets for building disease maps. It is complicated by technical batch effects on a sample level: cell type proportions change between datasets. Currently, no sample representation method addresses batch effects on the sample level. One possible direction would be to add a discriminator module to SampleCLR, which forces samples from different batches to mix with each other. Further research is required in this direction.

Given the broad assumptions of SampleCLR, it can be applied to other domains for set representation learning (Zaheer et al. (2017)). In biology, one possible extension would be cell painting data (Bray et al. (2016)), where images of thousands of cells are measured and then typically converted to tabular descriptions of cells. In single-cell transcriptomics, we envision that SampleCLR will pave the way to clinical applications of single-cell data.

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

# A ADDITIONAL THEORETICAL ANALYSIS

## A.1 SUBSAMPLING ROBUSTNESS: THEORETICAL GUARANTEE

The central biological assumption is that random subsamples of cells should give similar sample-level conclusions. We formalize this:

**Theorem A.1** (Subsampling Robustness). *Let $S_1, S_2$ be two random subsamples of size $m$ from sample $s$, with normalized importance weights $w_i^{(s)}$. Suppose $g_\phi$ is $L_g$-Lipschitz. Then*

$$\mathbb{E}[\|z(S_1) - z(S_2)\|_2^2] \leq \frac{2L_g^2}{m_{\text{eff}}}, \tag{7}$$

*where $m_{\text{eff}} = 1/\sum_i (w_i^{(s)})^2$ is the effective sample size.*

**Intuition**: When weights are uniform ($w_i = 1/n$), $m_{\text{eff}} = n$ and we recover the standard $\mathcal{O}(1/n)$ rate. When weights concentrate on few important cells, $m_{\text{eff}}$ is smaller, but those cells drive the phenotype signal.

**Connection to InfoNCE**: The contrastive loss directly minimizes $\mathbb{E}[\|z(S_1) - z(S_2)\|^2]$ for positive pairs, which is precisely the bound above. Negative pairs ensure separation across samples, tightening discriminability.

**Practical guidance**: In practice, stability holds for $m_{\text{eff}} \geq 10$. For rare-cell settings where $m_{\text{eff}}$ is small, oversampling or stratification can mitigate instability.

## A.2 COMPUTATIONAL COMPLEXITY

We analyze the asymptotic computational and memory cost of training and inference.

**Proposition A.2** (Computational Complexity). *Let $B$ denote the batch size, $m$ the feature dimensionality, and $N$ the number of test samples.*

- ***Training.*** *The pairwise similarity matrix requires $O(B^2 \cdot m)$ operations, since each of the $B^2$ pairs of embeddings involves an $m$-dimensional dot product or distance computation. Memory usage is $O(B^2)$ to store the similarity matrix, in addition to $O(B \cdot m)$ for the embeddings.*

- ***Inference.*** *Each sample is processed independently, yielding a cost of $O(N \cdot m)$ for $N$ test samples. Memory usage is $O(N \cdot m)$ if all embeddings are stored, or $O(m)$ if processed sequentially without caching.*

## A.3 FAILURE MODES AND PRACTICAL GUIDANCE

- **Rare cells**: If $m_{\text{eff}} < 10$, instability arises. Mitigation: stratified subsampling, targeted oversampling, or filtering out rare cell types.

- **Aggregator collapse**: when all cells get 0 weight, the model collapses. We found this to be an extremely rare case that can be mitigated by increasing the model size.

## A.4 IMPLEMENTATION NOTES

**Model architecture**

All the modules have the same underlying structure: input and output layers, transforming the data to correct dimensionality, and the series of hidden fully-connected layers of fixed dimensionality with residual connections. Each layer is followed by either batch or layer normalisation and ReLU activation function. We ran hyperparameter search with optuna, using validation loss as objective to find optimal set of hyperparameters. For datasets with less than 200 samples we use a "tiny" model:

```
{
    "num_layers": 4,
    "hidden_size": 32,
```

```
        "learning_rate_feature": 3e-4,
        "learning_rate_discriminator": 1e-5,
        "weight_decay": 1e-4,
        "n_aggregator_heads": 1,
        "aggregator_num_layers": 2,
        "aggregator_hidden_size": 32,
        "aggregator_activation": "relu",
        "output_dim": 32,
        "classifier_num_layers": 2,
        "classifier_hidden_size": 16,
        "regression_num_layers": 2,
        "regression_hidden_size": 16,
        "ordinal_num_layers": 2,
        "ordinal_hidden_size": 16,
        "use_normalization": false,
        "feature_normalization": "BatchNorm",
        "aggregator_normalization": "LayerNorm",
        "contrastive_loss": "XSampleCLR",
        "contrastive_loss_temperature": 0.1,
        "xsample_clr_graph_temperature": 0.2,
        "num_warmup_epochs_stage1": 20,
        "num_warmup_epochs_stage2": 10,
        "verbose": false,
        "early_stopping_patience": 150
}
```

For larger datasets, we use bigger version of the model:

```
{
        "num_layers": 16,
        "hidden_size": 128,
        "learning_rate_feature": 3e-5,
        "learning_rate_discriminator": 1e-5,
        "weight_decay": 1e-4,
        "n_aggregator_heads": 8,
        "aggregator_num_layers": 2,
        "aggregator_hidden_size": 128,
        "aggregator_activation": "relu",
        "output_dim": 192,
        "classifier_num_layers": 3,
        "classifier_hidden_size": 128,
        "regression_num_layers": 3,
        "regression_hidden_size": 128,
        "ordinal_num_layers": 3,
        "ordinal_hidden_size": 128,
        "use_normalization": false,
        "feature_normalization": "BatchNorm",
        "aggregator_normalization": "LayerNorm",
        "contrastive_loss": "XSampleCLR",
        "contrastive_loss_temperature": 0.1,
        "xsample_clr_graph_temperature": 0.1,
        "num_warmup_epochs_stage1": 20,
        "num_warmup_epochs_stage2": 10,
        "verbose": false,
        "early_stopping_patience": 150
}
```

If the model doesn't have supervised heads, another hidden layer is added to the network to equalise the number of parameters.

**Training protocol**: The model is trained in two stages: **Contrastive pre-training** and **Supervised fine-tuning**. During the first stage, only cell aggregator and encoder are trained to minimise contrastive loss, while in the second stage contrastive and supervised loss are used to update the entire model. We do a linear warmup for 10 epochs in the beginning of every stage and decrease learning

rate with a cosine scheduler towards the end of the stage. For the first 2 epochs of the 2nd stage the encoder and aggregator are frozen and only the supervised head is updated.

### A.5 CELL SAMPLING FOR INFERENCE

To remove bias related to cell number, we use a fixed number of cells for the final inference. While a random subset of cells can be used, akin to the training stage, a deterministic behaviour is desired. We noticed that removing cells with low eigenvector centrality based on nearest neighbor cell graph quickly deteriorates the representation quality of GloScope (Figure A1). Interestingly, subsampling random cells or cells with high eigenvector centrality does not lead to the same behaviour. Low eigenvector centrality values can therefore be used as a proxy for cell contribution to sample representation. We found this fact to be applicable to other datasets and metrics. We therefore use 1000 cells with the lowest eigenvector centrality to obtain sample representations with a trained SampleCLR model.

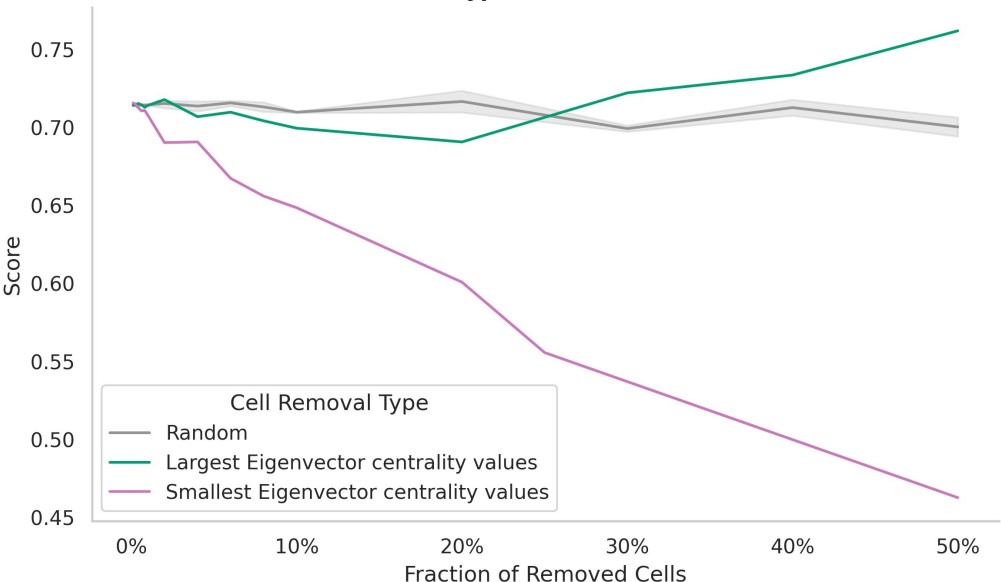

Figure A1: Low eigenvector centrality values are important for sample representation quality. The x-axis shows a percentage of removed cells for COMBAT dataset. The y-axis shows Spearman correlation for predicted disease outcome based on GloScope representation.

Eigenvector centrality is computed by solving

$$\lambda x^T = x^T A \tag{8}$$

in respect to $x$ associated with the largest positive eigenvalue $\lambda$. $A$ is the adjacency matrix of the nearest neighbour graph $G$, connecting each cell with $k$ most similar cells. We use $k = 15$ for all datasets.

### A.6 BENCHMARKING

#### A.6.1 DATASETS

We apply SampleCLR to 2 datasets of blood from healthy donors and COVID-19 patients: **COMBAT** (COvid-19 Multi-omics Blood ATlas (COMBAT) Consortium (2022)) and **Stephenson** (Stephenson et al. (2021)); dataset of blood from one thousand healthy donors **onek1k** (Yazar et al. (2022)), and human lung cell atlas extended with diseases (Sikkema et al. (2023)). Dataset characteristics and covariates used for evaluation are listed in table 2. We split each dataset into a training

and validation set. The former is used to train the supervised models, while the latter is only used for inference. We create validation set in a grouped way: putting entire batches, and in stratified way: making sure it has at least a uniform distribution for covariate used in a supervised task. Considering the low number of samples in the datasets, to let the model see enough samples, we limit validation set size to approximately 10% of samples per dataset.

Table 2: Datasets overview.

| Dataset | COMBAT | Stephenson | Onek1k | HLCA |
|---|---|---|---|---|
| #donors | 140 | 130 | **982** | 344 |
| #cells | 784k | 639k | 1.25M | **1.68M** |
| Tissue | PBMC | PBMC | PBMC | Lung and airways |
| Biologically relevant covariates | Disease, Severity, Death in 28 days, Duration | Disease, Severity, Outcome, Duration | Age | Tissue anatomical location, Disease, Smoking status |
| Technical covariates | Institute, Pool_ID | Site | Sex | Suspension type, Fresh or frozen, Sequencing platform, Assay |
| Trajectory covariate | Severity | Severity | Age | Anatomical location |
| Supervised task | Severity (ordinal regression) | Severity (ordinal regression) | Age (regression) | Disease (classification) |

### A.6.2 METRICS

We use the SPARE metrics (2.3 to evaluate the quality of the embeddings. Total score is obtained as weighted average of information retention, trajectory preservation and batch mixing scores with weight 0.5 for batch mixing and 1 for everything else. The score was then divided by 2.5 to make sure it is normalised from 0 to 1.

### A.6.3 METHODS

**Unsupervised baselines** We took the pre-calculated sample representations from the SPARE benchmark (Shitov et al. (2025)) provided by authors. To ensure that baselines are challenging, We took the best representation per method according to aggregated SPARE score.

**mcBERT** We ran the mcBERT model (Querfurth et al. (2024)) on a single Nvidia H100 GPU (80 GB of VRAM), 100 GB of system RAM, and four Intel Xeon platinum_8480+ CPUs. Each epoch took about 1.5 minutes. The workflow followed a two-stage pipeline, which includes pretraining with the Data2Vec class, then fine-tuning using supervised contrastive loss and inference to obtain patient embeddings. We used 3,000 highly variable genes (extended to a vocab size of 3,002 with CLS and MSK tokens). The model architecture had an embedding and hidden dimension of 288, 12 transformer layers, and 12 attention heads, containing a total of 26,564,256 trainable parameters. Pre-training was performed for 30 epochs with a batch size of 16, utilizing the Adam optimizer (learning rate = 1e-5, weight decay = 0.01), SmoothL1 loss ($\beta$ = 4) and EMA decay from 0.999 to 0. 9999 and Fine-tuning for 200 epochs (batch size = 32), with the same architectural settings, using supervised contrastive loss SupConLoss (temperature = 0.1) and early stopping (patience = 20). Both the pretraining and fine-tuning phases used oversampling of underrepresented disease categories and stratified train/validation/test splits based on a condition variable (e.g., disease type). Finally, Inference on the fine-tuned checkpoints to obtain patient-level embeddings and measured distances between samples using the cosine function.

**PaSCient training**:

PaSCient models were trained as multi-class classifiers using the architecture described in Liu et al. (2025). The input gene set comprised 3,000 genes selected during preprocessing, and the output layer size was dataset-specific ($n = 4$ for Stephenson, $n = 6$ for Combat, and $n = 8$ for HCLA), resulting in 4.9M trainable parameters. The Onek1k dataset with its regression tasks was excluded from model training and evaluation, as PaSCient is formulated for supervised classification tasks. Models were trained independently on each dataset using the designated sample holdout IDs, consistent with the procedures applied in the remaining methods. We used a batch size of batch_size = 16 with n_sampled_cells = 1000 cells per batch, using Adam optimizer (learning rate = 0.0002, weight decay = 0.0001). Oversampling was disabled. Training was performed for n_epochs = 4 on a single NVIDIA A100 GPU with 48 GB of memory, in line with the recommendations of the original PaSCient implementation.

**MultiMIL** MultiMIL models were trained according to the tutorial of the oficial package (Litinet-skaya et al. (2024)) with the recommended parameters. We were not able to run MultiMIL for onek1k dataset due to a bug in the current implementation of regression. Samples with missing task labels were removed. For classification and ordinal regression tasks, predictions were aggregated by majority voting across cells of each sample.

**MixMIL**

MixMIL (Engelmann et al. (2024)) was trained for 1000 epochs with the default parameters on the PCA representation of each dataset. We did not obtain results for onek1k dataset as the method does not support regression.

### A.7 ETHICS STATEMENT

We adhere to the ICLR code of ethics: https://iclr.cc/public/CodeOfEthics.

### A.8 LARGE LANGUAGE MODELS USAGE

Large language models (LLMs) were used to polish the text and to help with LaTeX formatting. No major pieces of text are generated by LLMs.

### A.9 REPRODUCIBILITY STATEMENT

SampleCLR code, dataset preprocessing, and benchmarking experiments are available on GitHub: https://github.com/sampleclr-iclr2026/SampleCLR. All the datasets in this study are publicly available. The authors will happily provide preprocessed datasets upon request.

