# OpenReview forum: "Great patients embed alike: contrastive learning for sample representation from single-cell data"
_ICLR.cc/2026/Conference — ICLR 2026 Conference Withdrawn Submission_

### Official Review · Reviewer_HZBS · 2025-10-30

**Soundness:** 2
**Presentation:** 2
**Contribution:** 2
**Rating:** 2
**Confidence:** 5

**Summary:**

The paper proposes SampleCLR, a contrastive learning framework for learning sample representations. The learned sample representations are benchmarked on the COVID-19 dataset in unsupervised and supervised settings.

**Strengths:**

- The motivation of the paper is clear, and the writing is generally easy to follow.
- The SampleCLR model uses fewer parameters compared to other sample aggregation methods.

**Weaknesses:**

The proposed subsampling strategy is straightforward, and the overall method lacks novelty. Even if the primary goal is to demonstrate the effectiveness of a SimCLR-like approach, the evaluation remains weak. No ablation studies are conducted on the key components of the SampleCLR model, and the experimental results are limited and not comprehensive.

**Questions:**

1. There are 4 metrics defined in Section 2.3, but why are there only three reported in Table 1?
2. How does the number of heads influence both the performance and interpretability of the SampleCLR model?
3. How do different subsampling ratios affect model training, and how should the optimal ratio be determined?
4. What is the performance of SampleCLR when trained directly without pre-trained weights? This would demonstrate the actual value of contrastive pretraining.
5. Since the four datasets involve different supervised tasks, how do all methods perform on each task individually?

---

### Official Review · Reviewer_spjc · 2025-10-31

**Soundness:** 2
**Presentation:** 2
**Contribution:** 2
**Rating:** 2
**Confidence:** 4

**Summary:**

This manuscript proposes SampleCLR, a contrastive self-supervised learning framework for generating patient-level (sample-level) embeddings from single-cell RNA-seq data. Instead of embedding individual cells, the authors treat each donor/sample as a distribution of cells and learn representations that remain stable under random subsampling. They leverage a contrastive objective where two subsamples from the same patient are treated as positive pairs, while subsamples from different patients form negatives. To improve interpretability and biological relevance, they introduce a cell importance weighting mechanism that learns which cell populations contribute most strongly to the sample representation. They also provide a supervised auxiliary head that can optionally fine-tune the shared embedding space for downstream clinical tasks. They benchmark the method with other baselines using metrics such as batching mixing, trajectory preservation, and information retention across COVID-19 PBMC cohorts and healthy donor atlas. SampleCLR outperforms or matches current state-of-the-art sample-level methods, while remaining parameter-efficient and yielding interpretable biological insights.

**Strengths:**

One of the key strengths of this work is its clear justification for operating at the patient level. Many clinically relevant phenotypes, such as disease severity or immune activation, ultimately manifest at the sample scale rather than the individual cell scale, and this distinction is framed convincingly. The contrastive learning strategy, based on consistency across random subsamples, is both elegant and intuitive, drawing a natural analogy to multi-view augmentations in computer vision. The inclusion of a learned cell-importance weighting adds interpretability and allows the model to highlight rare or transient cell states that could be overlooked by simple averaging. It is also appealing that the framework seamlessly bridges unsupervised representation learning with optional supervised refinement, allowing flexibility depending on label availability and noise. Empirically, the method performs strongly across several benchmarks while remaining computationally efficient compared to heavier baselines. Finally, the biological validation, showing that key immune programs relevant to COVID-19 severity emerge naturally, provides tangible evidence that the model captures meaningful signals rather than technical artifacts.

**Weaknesses:**

The manuscript has several areas that would benefit from clarification or additional support. First, the evaluation largely focuses on downstream benchmarking rather than carefully dissecting where the information comes from. It remains somewhat unclear whether the performance advantages are driven primarily by the contrastive objective, the pooling mechanism, the importance weighting, or some combination of these components. Second, the method assumes that random subsampling preserves the underlying patient state, but this assumption may break down in settings with extremely rare critical cell types or heterogeneous biopsy sampling, and the manuscript could more explicitly explore failure regimes. Third, while the importance weights are positioned as interpretable, their stability across replicates, cohorts, and model initializations is not fully demonstrated. Fourth, the optional supervised component raises the possibility of label leakage or overfitting when labels are sparse or noisy, yet there is limited discussion of regularization or safeguards. Finally, the biological validation focuses largely on one disease context, raising the question of whether the interpretability and performance generalize beyond the immunology context.

**Questions:**

1. Can the authors provide ablations that isolate the contributions of (a) contrastive subsampling, (b) importance weighting, and (c) multiple heads? This would clarify what drives the performance gains.
2. How stable are the learned importance weights across different training seeds or across biological replicates? Without stability, interpretability claims may be fragile.
3. In diseases where extremely rare cell types carry crucial signals (e.g., minimal residual disease, tumor-reactive T cells), does random subsampling introduce a risk of losing signal?
4. How does SampleCLR perform when tissue sampling depth or dissociation efficiency varies substantially between patients?
5. Since clinical labels are often noisy, what steps prevent overfitting when supervised heads are used?
6. Most shown examples involve blood- or immune-related contexts. Would the method perform similarly in spatially heterogeneous tissues such as tumors or brain?
7. Can importance weighting distinguish subtle transcriptional programs within the same nominal cell type, or is the attention largely modality-wide?
8. How does training time scale with increasing cell counts or expanded patient cohorts? Are there practical limits?

---

### Official Review · Reviewer_bHuA · 2025-10-31

**Soundness:** 2
**Presentation:** 2
**Contribution:** 2
**Rating:** 2
**Confidence:** 5

**Summary:**

In this paper, the authors proposed SimCLR, a contrastive learning based method aim at learning the representations of a population of cells (sample level representation learning), and performed 2 layers of benchmarking experiments against supervised learning based and unsupervised learning based methods. They also claimed that the cell aggregator module embedded in the model can provide a certain level of interpretation ability at cellular scale.

**Strengths:**

1. The paper is generally well written.

**Weaknesses:**

1. What's the hint of learning sample level representations? Suppose we take all single cells embeddings from foundation models like scGPT/geneformer, and use the 2d matrix as the "sample level representation", in which task and how will the single patient vector outperform the 2d matrix?? Moreover, if we simply do a linear projection of the 2d matrix -- even do a simple "pseudo-bulk" on top of the embedding space -- it will be a powerful sample level representations. Then, why do we need contrastive learning for extracting the features from the raw count matrix?
2. Lack of discussion about approaches the reviewer proposed in 1.
3. In section 2.3, I did not see much differences of the sample level benchmark metrics against single cell level metrics. If the sample level task is biologically meaningful, one should propose more novel and biological problem related metrics. Current metrics are far from enough.
4. Moreover, some very basic metrics, like reverse single cell level summary statistics recovery (e.g. pseudo bulk expression; gene expression variance across samples) are not proposed.
5. Current pseudobulk baseline is very weak. A more suitable pseudobulk is the pseudobulk/linear projection in the representation space for a population of cells. (E.g.,  PCA/scVI/GeneFormer/scGPT )
6. Some prior contrastive learning model on single cell omics, like CLEAR and contrastiveVI, are not discussed.

**Questions:**

See weakness above.

---

### Official Review · Reviewer_B1s8 · 2025-10-31

**Soundness:** 3
**Presentation:** 2
**Contribution:** 2
**Rating:** 4
**Confidence:** 4

**Summary:**

This paper proposes a method to do sample-level representation learning of single cell data. The method works using a contrastive approach, defining a target similarity matrix based on augmentations (resampled version of a single cell sample) and a similarity prior extracted from another model. The model then use a X-constrastive loss.

The method can optionally be made supervised by adding a classifier or regression head that can be trained jointly.

The authors evaluate their approach on SPARE, a dedicated benchmark for sample level representation learning. The results show that the GloScope prior helps in the supervised setting but not in the unsupervised case.

The authors also interpret the aggregation weights learnt by their method and claim that it can be used to learn new biological patterns.

**Strengths:**

- Sample level representation of single cell data is an important problem that can unlock at lot of impactful computational biology applications.
- The method makes sense, and is well suited given the problem formulation.
- The authors use standardized benchmarks for evaluation

**Weaknesses:**

- Critical details of the methods description are missing from the paper. It's not clear how the target similarity matrix is computed in their X-CLR framework (how are the prior and the augmentations combined ? ). It's also not fully clear what the final loss is in the supervised case, it seems that the X-CLR loss is always used but would be great if the text could confirm it.
- Pseudobulk baselines should be included in the supervised methods. SPARE evaluates information retention with a kNN. A simple baseline is to train an MLP from pseudo-bulk to predict the label and use the last layer activation as the embedding. Pseudobulk representations have shown to recapitulate a lot of information, but using raw pseudobulk will suffer from high-dimensionalty - which will impact the kNN performance. You can also do the same with cell type composition. This is an important experimental detail that should appear in the main text.
- SampleCLR is primarily presented as an unsupervised method. However, it is outperformed by very simple baselines like GloScope. It's not clear one would prefer SampleCLR over something like GloScope in practice.
- The machine learning contribution of this paper is limited. The subsampling augmentation is not really studied extensively and SampleCLR is somewhat a direct application of X-CLR.

**Questions:**

- Could the authors please expand on the experimental details (as requested above) ?
- Could the authors add the pseudobulk and cell type proportion baselines in a supervised setting ?
- SPARE contains 5 internal datasets but I could not find the breakdown of the results in the paper. Could you please add the detailed results in the appendix ?
- Could the authors explain why someone would use SampleCLR rather than GloScope in practice given the results ?

---

### Note · Authors · 2025-11-24

**Comment:**

We thank the reviewers for their feedback. Overall, the reviewers appreciated our methodology and suggested relevant improvements. While we think we can address these, we would like to withdraw to take more time to improve the paper before targeting a different venue. We thank reviewers and AC for their time.

Additionally, we would like to raise concerns that one of the reviews is classified as 100% AI-generated: https://iclr.pangram.com/reviews?submission_number=22370. In that light, we especially appreciate reviewers who provided the feedback themselves.

**Withdrawal Confirmation:**

I have read and agree with the venue's withdrawal policy on behalf of myself and my co-authors.